# Melanoma Antigen Family A (MAGE A) as Promising Biomarkers and Therapeutic Targets in Bladder Cancer

**DOI:** 10.3390/cancers16020246

**Published:** 2024-01-05

**Authors:** Shiv Verma, Diya Swain, Prem Prakash Kushwaha, Smit Brahmbhatt, Karishma Gupta, Debasish Sundi, Sanjay Gupta

**Affiliations:** 1Department of Urology, School of Medicine, Case Western Reserve University, Cleveland, OH 44106, USA; sxv304@case.edu (S.V.); ppk22@case.edu (P.P.K.); kxg126@case.edu (K.G.); 2The Urology Institute, University Hospitals Cleveland Medical Center, Cleveland, OH 44106, USA; 3College of Arts and Sciences, Case Western Reserve University, Cleveland, OH 44106, USA; dxs1086@case.edu (D.S.); ssb120@case.edu (S.B.); 4Department of Urology, Division of Urologic Oncology, The Ohio State University Comprehensive Cancer Center, James Cancer Hospital & Wexner Medical Center, Columbus, OH 43210, USA; d.sundi@osumc.edu; 5Department of Pathology, Case Western Reserve University, Cleveland, OH 44106, USA; 6Department of Pharmacology, Case Western Reserve University, Cleveland, OH 44106, USA; 7Department of Nutrition, Case Western Reserve University, Cleveland, OH 44106, USA; 8Division of General Medical Sciences, Case Comprehensive Cancer Center, Cleveland, OH 44106, USA

**Keywords:** bladder cancer, Melanoma Antigen Gene, cancer testis antigen, biomarkers, therapeutic target, signaling pathways, protein–protein interaction, genomic alterations

## Abstract

**Simple Summary:**

The Melanoma Antigen Gene (MAGE) belongs to the larger family of cancer testis antigens. The MAGEA family were the first tumor-associated antigens identified at the molecular level whose expression was consistent in most human cancers and germinal cells. Aberrant expression of MAGEA family is noted in a majority of human malignancies, where they are associated with increased cancer cell proliferation, survival, and resistance to various therapies. This makes them an ideal biomarker and attractive therapeutic target in designing novel therapies. This review mainly focuses on the opportunities for the development of MAGEAs as promising biomarkers and their therapeutic implications in bladder cancer.

**Abstract:**

The Melanoma Antigen Gene (MAGE) is a large family of highly conserved proteins that share a common MAGE homology domain. Interestingly, many MAGE family members exhibit restricted expression in reproductive tissues but are abnormally expressed in various human malignancies, including bladder cancer, which is a common urinary malignancy associated with high morbidity and mortality rates. The recent literature suggests a more prominent role for MAGEA family members in driving bladder tumorigenesis. This review highlights the role of MAGEA proteins, the potential for them to serve as diagnostic or prognostic biomarker(s), and as therapeutic targets for bladder cancer.

## 1. Introduction

Bladder cancer continues to be an important health problem with an estimated 600,000 new cases and 220,000 deaths worldwide [1]. In the United States, more than 80,000 new cases and 17,000 deaths are predicted this year [2]. Urothelial carcinoma is the most common type of bladder cancer, which arises from the urothelial cells lining the bladder’s inner surface. At presentation, nearly 75% of patients have non-muscle-invasive bladder (NMIBC) cancer and 25% have muscle-invasive (MIBC) disease. About 50% of NMIBC are low-grade, whereas most muscle-invasive or metastatic tumors are high-grade and invade the detrusor muscle [2,3,4]. NMIBC is typically managed via cystoscopic transurethral resection in combination with adjuvant intravesical chemo- or immune-therapy [2,5]. Unfortunately, disease recurrence is common (40–75%) despite guideline-concordant care, and approximately 30% of cases progress to MIBC, which often requires systemic chemotherapy with radical pelvic surgery (cystectomy) or chemo-radiation [6,7]. Treatment recommendations are based on risk stratification, which presently do not include biological heterogeneity in bladder cancer. One way to address this gap is to identify novel biomarkers and therapeutic targets to manage this life-threatening disease.

In the past decade, the search for diverse biomarkers involved in cancer initiation and/or its progression led to the discovery of cancer–testis antigens (CTAs) as a breakthrough in cancer biology and clinical oncology [8,9,10]. CTAs are commonly expressed in various tumor types but have limited expression patterns in normal tissues; therefore, they are proposed as cancer biomarkers for diagnosis [11,12,13,14,15]. Some of these molecules are progressively increased during carcinogenesis and thus recognized as prognostic markers; both could be effective targets for prevention and/or therapeutic intervention. In the human genome, more than 200 CTA genes have been documented and classified into 44 gene families in the CTA database (http://www.cta.lncc.br (accessed on 4 September 2023)) and GeneBank (https://www.ncbi.nlm.nih.gov/gene (accessed on 5 September 2023)) [16,17]. A few of these families consist of multiple members and CTA gene orthologs and paralogs. CTAs are divided into two groups based on the chromosomal location that include CT-X antigens positioned in the X chromosome or non-X CTAs sited on the autosomes [18]. The distribution of antigens on the X-chromosome (Xq21-q28 region) is organized in gene clusters harboring groups of direct and inverted repeats, whereas non-X-CTA genes are mostly single-copy genes [19]. The expression of type I or CT-X is restricted to the X chromosome that comprises three subfamilies, MAGEA, MAGEB, and MAGEC, whereas type II, which is not limited to the X-chromosome, includes MAGED, MAGEE, MAGEF, MAGEG, MAGEH, MAGEL, and Necdin [20]. Additionally, CTA pseudo-genes were identified in the MAGEA, MAGEB, SSX, CT45, and CT47 families in human and mammalian genomes [21]. The expression of 174 CTA-encoding genes might be regulated by CTA-non-coding RNAs, which are annotated in the Cancer Genome Atlas (https://cancergenome.nih.gov (assessed on 6 October 2023) [22]. However, both type I and type II groups have a Melanoma Antigen family (MAGE) homology domain (MHD), which is highly conserved within the MAGEA subfamily (>80% identical) consisting of approximately 170 amino acids, except the MAGED proteins, which consist of two MHDs [23,24]. Structure analysis has documented that MHD is a tandem-winged helix motif that presumably plays a key role in protein–protein interaction [25]. While other studies have demonstrated the similarity between MAGE proteins having distinct functions [26], it was projected that the adaptable MHD undergoes allosteric changes to allow interactions between different protein domains, conferring special properties to MAGE members [27,28]. The members of MAGED family were considered the ancestral and most conserved with the highest homology of gene and protein sequences between humans and other species based on alignment score data (https://www.ncbi.nlm.nih.gov/homologene (accessed on 16 October 2023) (Figure 1).

Members of the MAGEA family were the first tumor-associated antigens identified in humans at the molecular level [29,30,31]. To date, 12 family members of MAGEA have been identified, including type I MAGEs, that are characteristically restricted to expression in the testis and are often abnormally expressed in various human malignancies. Preferential intracellular location may be dissimilar for different antigens, such as MAGEA1, MAGEA3, and MAGEA4, which are mostly cytoplasmic, but MAGEA10 is majorly nuclear in localization [32,33]. The mechanisms that control the unusual re-expression of MAGEAs are still under exploration. MAGEAs are infrequently expressed in somatic tissues; however, epigenetic changes including DNA hypermethylation of CpG dinucleotides in promoters and posttranslational modifications of the histone proteins averts transcription factors from binding, and thus represses the expression of MAGEA genes [34]. Further studies have shown that DNA methyltransferases (DNMTs) inhibitors, such as 5-aza-2-deoxycytidine, result in the increased expression of MAGEA1 in several malignant cells [35]. This effect can be additionally reinforced using histone deacetylase inhibitors [35,36].

Several studies demonstrated the role of MAGEA family members in cancer cell proliferation and progression [27,37,38]. The aberrant expression of MAGEA3 facilitates cervical cancer proliferation and metastasis involving the Wnt signaling pathway [39]. The overexpression of MAGEA4 facilitates the growth of spontaneously transformed oral keratinocytes through apoptosis inhibition [40]. Other studies have found that MAGEA4 increases the survival of cancer cells through its interaction with Gankyrin and Miz1, which are transcriptional partners of c-myc. The recruitment of the MAGEA4-Miz1 transcriptional complex on the Cip1/p21 promoter results in the downregulation of Cip1/p21, thus enhancing cancer cell survival [41]. Through direct interaction, MAGEA3 and MAGEA6 are reported to be involved in the ubiquitination of AMPKα 1 catalytic subunit regulating autophagy and adaptation to nutrition stress [42]. This event results in the upregulation of mTOR signaling pathways facilitating early tumor formation [43]. Furthermore, MAGEA11 has been shown to interact with S phase kinase-associated protein (Skp2) modulating its specificity and association with cyclin A regulating cell cycle progression [44].

Numerous studies detail the expression pattern of MAGEA members in various human tumors, while only few studies have identified mutations across different tumor types. A somatic mutation analysis of the coding regions of MAGEA family showed mutations in one or more members in melanoma patients [45]. An analysis of MAGEA4 mutant proteins in tumor cells exhibited high structural stability but showed changes in thermal stability and folding that affect tumor growth [46]. Based on the mutation analysis, an overall low level of correlation was detected between MAGEAs mutation and antigen expression that might contribute to tumor progression. 

Clinical studies have shown a correlation between MAGEA expression and poor prognosis in cancer patients [47]. Studies also suggest that MAGEA protein expression is associated with therapeutic resistance [48,49]. There is increasing evidence demonstrating the involvement of MAGEA proteins in regulating the processes of cell survival in cancer cells by direct interaction with the p53 tumor suppressor or indirectly by regulating the activity of E3 RING ubiquitin ligases [50,51]. MAGEA proteins increase the metastatic potential of malignant cells by enhancing cell motility and invasiveness [47,52]. Overall, MAGEA antigens have been investigated for their role in human cancers and as potential therapeutic targets. This review highlights the role of MAGEA family proteins in bladder cancer as putative diagnostic/prognostic biomarkers and in future directions toward advancements in MAGEA-specific therapies.

## 2. Expression of MAGEA Family in Bladder Cancer

There are a few reports on the expression of specific MAGEA family members in bladder cancers. Patard et al. (1995) analyzed the expression of MAGEA1, MAGEA2, MAGEA3, and MAGEA4 and observed the expression of at least one of these genes in 61% of invasive and 28% of superficial tumors, with MAGEA3 and MAGEA4 genes being most frequently expressed [53]. A study by Picard et al. (2007) analyzed the expression of MAGEA3, MAGEA4, MAGEA8, and MAGEA9 in bladder cancer. The study reported that MAGEA3–9 members were expressed in 30%, 33%, 56%, and 54% of bladder tumors, respectively. Although MAGEA8 was the most frequently expressed, its expression was low overall and mostly confined to the normal urothelium. In comparison, MAGEA9 was expressed at a higher level and was two times more frequent in superficial bladder cancer than in invasive tumors [54]. Bergeron et al. (2009) showed that MAGEA4 and MAGEA9 were expressed in 38% and 63% of NMIBC, in 48% and 57% of MIBC, 65% and 84% in carcinomas in situ, and 73% and 85% in lymph node metastases, respectively. The expression of MAGEA4 (*p* = 0.007) and MAGEA9 (*p* = 0.012) was associated with higher-grade tumors. In multivariable Cox regression analyses, the expression of MAGEA9 in pTa tumors was associated with recurrence (HR = 1.829; *p* = 0.010). MAGEA4 expression in these tumors was associated with progression to MIBC (HR = 7.417, *p* = 0.013) based on univariate analyses, whereas MAGEA9 expression was further predictive of bladder cancer progression [55]. A study by Xylinas et al. (2014) showed that MAGEA3 expression was independently associated with an increased risk of bladder cancer recurrence and cancer-specific mortality [56]. Another study by Dyrskjot et al. (2012) showed that 43% of bladder tumors expressed MAGEA3. A univariate Cox regression analysis of gene expression in NMIBC showed that the expression of MAGEA3 (*p* = 0.026) was significantly associated with a shorter progression-free survival [57]. A study by Kocher et al. (2002) showed that MAGEA4 protein was significantly expressed at higher levels in transitional cell carcinomas (*p* < 0.001); its positivity was significantly correlated with an invasive phenotype (*p* < 0.001) and high-grade tumors (*p* < 0.0001). A retrospective evaluation of 908 transitional cell carcinomas of the bladder patients demonstrated strong MAGEA4 staining which was associated with decreased tumor-specific survival (*p* < 0.0001) [58]. Other studies have shown higher expressions of MAGEA2, MAGEA8, and MAGEA10 in high-grade bladder tumors [59,60,61,62,63]. Overall, these studies suggest that MAGEA members are associated with bladder cancer, grade, stage, and oncologic outcomes.

We performed a meta-analysis combining the results from several independent studies on bladder cancer listed in the TCGA database for detecting differentially expressed MAGEA genes with the potential to increase both the statistical power and generalizability of our analysis. MAGEA family expression was analyzed in n = 4536 patients according to tumor stage and the *p* values were generated by two-sample t-test, comparing pTa and pT2 stage bladder cancer. MAGEA2 (*p* < 0.001), MAGEA3 (*p* = 0.005), MAGEA6 (*p* < 0.001), and MAGEA12 (*p* = 0.01) were found to have stage-dependent expression (highest in pT2 subgroup), suggesting that these MAGEA members might play important roles in the development of urothelial carcinoma (Figure 2). These observations highlight the importance of improving our understanding of the etiology of bladder cancer, as well as the molecular changes underlying aberrant MAGEA expression. However, the clinical and prognostic value of MAGEA family members in the pathobiology of bladder cancer is currently under investigation by our group.

## 3. Genomic Aberration of MAGEA Genes in Bladder Cancer

To gain a better understanding of the molecular alteration of MAGEA proteins in bladder cancer, we used a TCGA bladder cancer patient’s (n = 4536) genomics database to analyze how MAGEA family members could affect bladder tumorigenesis. Genomic alterations, including mutation, homozygous deletion, or amplification, led to uncontrolled proliferation and irregularities in cell death in neoplastic cells [64,65]. It was found that frequencies of genomic alterations among MAGEA family members in bladder cancer were MAGEA1 1.8%, MAGEA2 1.6%, MAGEA3 1.9%, MAGEA4 1.6%, MAGEA6 2.6%, MAGEA8 1.7%, MAGEA10 2%, MAGEA11, 2.1%, and MAGEA12 2.4% (Figure 3A). The most prevalent form of genomic alteration is gene amplification among all MAGEA members [66,67]. The amplification of these genes often has the potential to transform normal cells into neoplastic cells, further hinting at the possibility that the MAGEA family has a significant role in bladder tumorigenesis [68]. These observations recommend that an analysis of MAGEAs might be a useful diagnostic tool to determine the invasive potential of bladder cancer. In addition, the amplified regions of some MAGEA genes might serve as potential therapeutic targets.

We also analyzed mutations in the MAGEA family in bladder cancer. MAGEA genes are more frequently mutated in cancer patients, suggesting their critical role in malignancy [69]. In bladder cancer, the somatic mutation frequency is very low and ranges from 0.1% to 0.4% in the MAGEA family members identified in 4880 samples from 4158 patients consolidated from 21 studies from the cancer genome atlas (TCGA) (Figure 3B). 

Moreover, missense mutations were identified in MAGEA6 compared to MAGEA3 and MAGEA11. Some of these mutations, including G137W, E232Q, P242L, Y249H, P262R, G296V, R298C, and E314Q, were found in the 229–399 amino acid sequence in the MAGEA6 gene [56]. Among the noted mutations, the functional impact of P262R was most deleterious, which might contribute to poor outcomes in bladder cancer patients (Figure 4A,B).

## 4. Gene Network and Signaling Pathways of MAGEA Family in Bladder Cancer

The altered expression of genes results in changes in gene expression and gene network interaction during cancer progression. A Cytoscape version 3.10.1 (Complex Network Analysis) Genemania module [70] was used to explore the genetic interaction of MAGEA1, MAGEA2, MAGEA3, MAGEA4, MAGEA6, MAGEA8, MAGEA10, MAGEA11, and MAGEA12 (red color). This software tool provides a critical assessment and integration of protein–protein interactions to assess the associations of potential differentially expressed genes. The size of the circle of each protein represents its degree of connection to other proteins. The analysis demonstrated the interaction of MAGEA family members with other MAGE members, including MAGEA2B, MAGEB10, MAGEB2, and others (blue color). The available scientific literature reports on the interaction between MAGEA and MAGEB in bladder cancer [71]. Moreover, both family members share the MAGE domain that influences the tumor microenvironment and promotes cell proliferation (Figure 5A).

We further assessed the alteration in the MAGEA family of proteins affecting various signaling pathways promoting tumorigenesis [53,54,55,56,57,58,59,60,61,62,63]. We analyzed three independent studies on bladder cancer (GSE154261, GSE57813, and GSE37317) using Ingenuity Pathway Analysis (IPA). Our analysis revealed that the molecular mechanism of cancer, mitochondrial dysfunction, protein ubiquitination, oxidative phosphorylation, and sirtuin signaling pathways are among the top five signaling networks associated with MAGEAs expression in bladder cancer. Furthermore, it was documented that a high expression of MAGEA3 modulates the function of the AMPK pathway and significantly decreases autophagy, leading to the activation of the mTOR signaling pathway [72] (Figure 5B).

## 5. MAGEA Family as Diagnostic Biomarkers in Bladder Cancer

Several studies have analyzed the stage-specific expression of the MAGEA family in bladder cancer, indicating their higher levels in invasive disease [53,54,55,56,57,58,59,60,61,62,63]. Therefore, we explored the TCGA database to determine the expression of different MAGEA members in bladder cancer (BCLA; n = 414) compared with normal bladder samples (n = 19). The BCLA patient’s dataset was not available for MAGEA9, and MAGEA5 and MAGEA7 were excluded from the analysis as they are pseudogenes. In some databases, the expression of MAGEA5 and MAGEA7 are measured, despite being pseudogenes, as they regulate oncogenes by serving as miRNA decoys [73]. Interestingly, MAGEA2, MAGEA3, MAGEA4, MAGEA6, MAGEA10, MAGEA11, and MAGEA12 exhibited significant differences in their expression in bladder cancer compared to normal bladder samples. Other MAGEA family members, including MAGEA1 and MAGEA8, exhibited lower expression in bladder cancer compared to normal bladder specimens (Figure 6). A data analysis of bladder cancer patients further uncovers the possibilities of utilizing MAGEA2, MAGEA3, MAGEA4, MAGEA6, MAGEA10, MAGEA11, and MAGEA12 members as diagnostic biomarkers for bladder cancer.

Next, we determined the cancer-specific mortality using a hazard ratio (HR) of MAGEA proteins with overall survival for the assessment of relative risk for bladder cancer aggressiveness, using the GENT2 database, which associates gene expression with the HR. The HR was calculated using the fixed effect model and the random effect model with 95% CI and *p*-value. Based on the data analysis from various studies, the HR value for the fixed (FE) and random effect (RE) models were as follows: MAGEA1 (FE, 1.06; RE 1.18; *p*-value = 0.07); MAGEA2 (FE, 1.48; RE, 1.48; *p*-value = 0.67); MAGEA3 (FE,1.43; RE, 1.43; *p*-value = 0.33); MAGEA4 (FE, 0.96; RE, 1.21; *p*-value = 0.02); MAGEA6 (FE, 0.99, RE, 1.05; *p*-value = 0.03); MAGEA8 (FE, 1.05, RE, 1.05; *p*-value = 0.86); MAGEA10 (FE, 1.07.; RE,1.10; *p*-value = 0.11); MAGEA11 (FE, 1.09; RE,0.94; *p*-value = 0.31); and MAGEA12 (FE, 1.00; RE, 1.73; *p*-value < 0.01). The HR ratio of MAGEA6 and MAGEA12 was statistically significant with *p*-value = 0.03; <0.01. Based on these findings, poor survival might be predicted in patients that express elevated levels of MAGEA6 and MAGEA12. To define the variation among different datasets, the heterogeneity was calculated. Heterogeneity was common regardless of the treatment effects by odds ratios or risk differences. Random effects estimates, which incorporate heterogeneity, tended to be less precisely assessed than fixed effects estimates. Therefore, compared with the fixed effect model, the weights assigned under random effects are more balanced. The heterogeneity was 0%; this means that heterogeneity has no importance in the results displayed in the forest plot (Figure 7).

## 6. Prognostic Value of MAGEA Gene Family in Bladder Cancer

We analyzed the prognostic value (log rank) of MAGEA family members viz. MAGEA -1, -3, -4, -6, -8, -10, -11, and -12 in bladder cancer patients (n = 408; MAGEA2, n = 406) using KM-Plot (http://kmplot.com/analysis (accessed on 19 October 2023)). For this, we performed survival analysis in bladder cancer patients using the TCGA database. A hazard ratio (HR) of more than 1.0 was used to predict the potential of genes as prognostic biomarkers. Based on the log rank value of MAGEA family members, we found that MAGEA6 scored the highest log rank (log rank 0.99), followed by MAGEA3 (log rank 0.83), compared to other MAGEA members. These values suggest that MAGEA3 and MAGEA6 may serve as prognostic biomarkers for bladder cancer patients (Figure 8). Additional clinical validation in bladder cancer patients is required to justify the above rationale.

Next, we determined the predicted location and protein–protein interaction of MAGEA3 and MAGEA6 genes in bladder cancer. MAGEA6 protein is intracellular in location and showed direct interaction (physical association) with secreted (intracellular) protein S100A9, CTF1, GFOD1, and APO4. MAGEA6 showed interaction with intracellular proteins such as TULP3, EXOC5, LSM2, and others. Similarly, MAGEA3 showed interaction with secreted intracellular protein S100A9, along with other intracellular proteins. Genomic association revealed a direct correlation of MAGEA3 and MAGEA6 with calcium-binding protein S100A9 (Figure 9A,B). To further understand the association between MAGEA3 and MAGEA6 with S100A9, we explored the OncoDB cancer database, a large-scale multi-omics database, and performed pairwise gene expression correlation analysis between MAGEA3 and MAGEA6 and S100A9 in bladder cancer patients (Figure 10). S100A9 is a secretory protein, and its expression is positively associated with elevated levels of MAGEA3 and MAGEA6 proteins [74,75]. An overexpression of S100A9 is associated with stage progression, invasion, metastasis, and poor survival in bladder cancer patients [76]. S100A9 and MAGEA family members may be cooperative oncogenes, given our findings. 

## 7. MAGEA Family as Therapeutic Target in Bladder Cancer

The basic potential strategies for MAGEA-targeted therapy in bladder cancer include immunotherapy against MAGE epitopes, the interruption of MAGEA–partner interactions, and the manipulation of regulatory pathway(s) affecting MAGE function. The role of MAGEA proteins had been established and demonstrated as therapeutic targets in multiple myeloma (MM) [77,78] and esophageal cancer [79]. The first two approaches might have huge benefits because of the limited expression of MAGE proteins in normal tissues (https://www.proteinatlas.org (accessed on 19 October 2023). The development of small molecule inhibitors interacting with MAGEA protein could have a lesser effect on somatic tissues and therefore minimize side effects [50,80,81,82]. The immune-based approaches are preferable due to the strong natural immunogenicity of MAGEA proteins coupled with the fact that germ cells do not express MHC class I antigens [30,83,84,85]. Hence, MAGEA-targeted vaccines should not elicit an autoimmune response in the testis. To date, there are approximately 47 oncologic clinical trials (including five involving bladder cancer) with a major focus on MAGE proteins. The clinical trials focusing on MAGEA proteins in bladder cancer are summarized in Table 1.

Advancements in understanding the biology of MAGEA proteins are still ongoing; in particular, the synthesis of stable peptide inhibitors is a major breakthrough in the use of MAGEA–protein interactions [92,93,94]. The hydrocarbon cross-linker stable peptides can be synthesized for targeting surface proteins, protease resistance, and cell permeability [95]. There is also a growing list of small molecules that have been shown to be effective in inhibiting protein–protein interactions, including MAGEA function. For example, a study by Bhatia et al. (2011) targeted the interaction between the RBCC domain of KAP1 with MAGE proteins [96].

MAGEA proteins were the first human tumor-associated antigens identified at the molecular level. These proteins are more frequently expressed in the majority of tumors and are recognized as more potent and promising immunotherapeutic targets. The expression of type I MAGEs is typically restricted to the testis except in various cancers, where abnormal expression defined the term cancer–testis antigen. It is not just genomic dysregulation that induces the aberrant expression of MAGEs; interestingly, MAGEs contribute actively to tumorigenesis. Although several studies have associated MAGEs (including MAGEA2, MAGEA3, MAGEA6, and MAGEA9 expression) with pro-tumorigenic activities, such as p53 dysregulation, enhanced tumor proliferation, or the maintenance of cancer-stem-cell-like characteristics [97,98], their definitive functions are not still fully understood in the context of bladder cancer.

## 8. Conclusions and Future Directions

An examination and study of the human bladder cancer database has uncovered the MAGEA family and identified its diverse cellular functions, though this is just the beginning stages of understanding how the MAGEA family contributes to normal physiological processes versus the pathogenesis of bladder cancer. As highlighted in this review, the in-depth genomic alteration, molecular structure, genetic interaction, and signaling cascades of the MAGEA family will provide further insight into its molecular role during bladder cancer pathogenesis. MAGEA genes can promote tumor progression through various mechanisms, such as through the activation of androgen receptor (AR), p53 inactivation, and an increase in oxidative phosphorylation (OXPHOS), which is overrepresented in bladder cancer and eventually contributes to highly aggressive and metastatic disease states in bladder cancer patients. Therefore, the MAGEA family has been considered as potential targets for bladder cancer treatment.

The high expression of MAGEA family members in bladder cancer may increase the propensity of proliferation and invasiveness during disease progression. This relationship makes MAGEA family members potential targets for diagnostic biomarkers, especially for muscle-invasive bladder cancer. Due to their highly antigenic properties and other diverse roles during cancer progression, the aberrant expression of MAGEA family, especially MAGEA3 and MAGEA6, may serve as prognostic biomarkers for poor outcomes in bladder cancer patients. Based on the genomic alteration profile of MAGEA proteins, it was speculated that mutation in the protein P262R may change the protein structural configuration which could attract other hormones and proteins facilitating malignant progression. MAGEA proteins may serve as a hormone receptor coregulators and trigger the pathogenic response. More mechanistic studies of the MAGEA family will facilitate the development of targeted therapies.

MAGEA-targeted approaches could be effective in inhibiting or even eliminating MAGEA cancer-promoting activities in various human cancers. Based on the current understanding of MAGEA proteins and their expression in germ cells, targeted therapy may offer a high degree of specificity with minimal side-effects in clinical settings. Thus, MAGEA family seems to be highly attractive area of research based on recent advancements in molecular biology. Additionally, innovative technologies are now available to enhance the immunotherapeutic responses and to target protein–protein interactions, which may be relevant for MAGEA proteins. Therefore, insights into the role of MAGEA proteins at the molecular level, and their association with cancer, should provide exciting new opportunities for exploitation and therapeutic intervention. In summary, this review emphasizes the importance of MAGEA family proteins in bladder cancer.

## Figures and Tables

**Figure 1 cancers-16-00246-f001:**
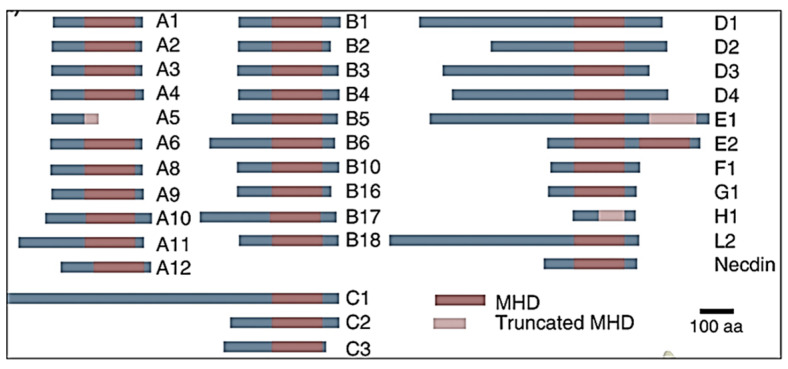
Human MAGE proteins with the identified common domain, the MAGE homology domain (MHD). Few MAGEs have truncated MHDs and those members that are likely pseudogenes are not listed.

**Figure 2 cancers-16-00246-f002:**
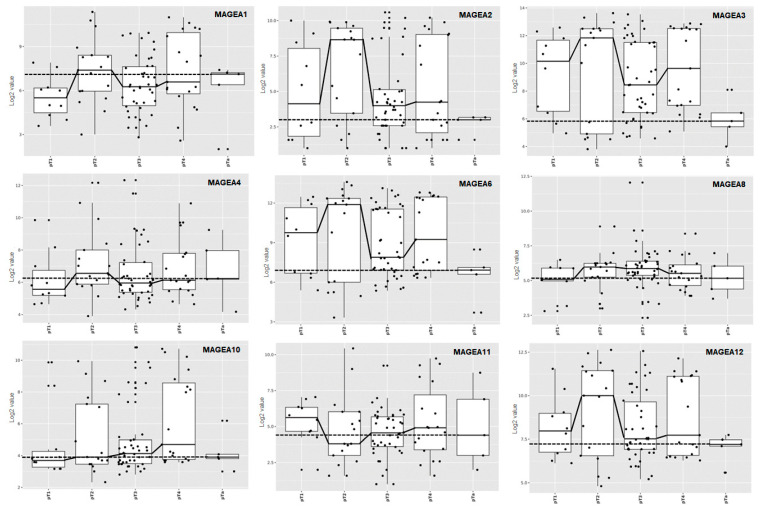
MAGEA expression profile across various stages of bladder cancer. Box and dot-plot of bladder cancer stages. The X-axis of the graph represents various stages of bladder cancer, pT1, pT2, pT3, pT4, and pTa, and the Y-axis of the graph represents log2 value.

**Figure 3 cancers-16-00246-f003:**
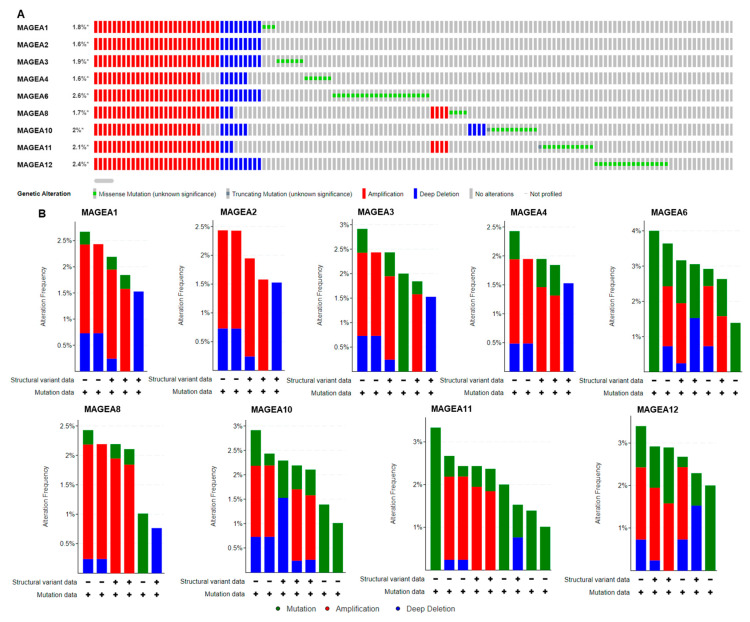
Genomic alterations in MAGEA family members. Oncoprint of MAGEA1, MAGEA2, MAGEA3, MAGEA4, MAGEA6, MAGEA8, MAGEA10, MAGEA11, and MAGEA12 in bladder cancer patients (n = 5436). (**A**) Color bar represents the individual patient’s profile showing gene amplification in red, * denotes % gene amplification; deletion in blue; and mutation in green. (**B**) Bar graph represents the summary of MAGEA gene alteration frequency (Y-axis), and the X-axis shows the mutation rate (green), amplification (red), and deep deletion (blue) in bladder cancer patients.

**Figure 4 cancers-16-00246-f004:**
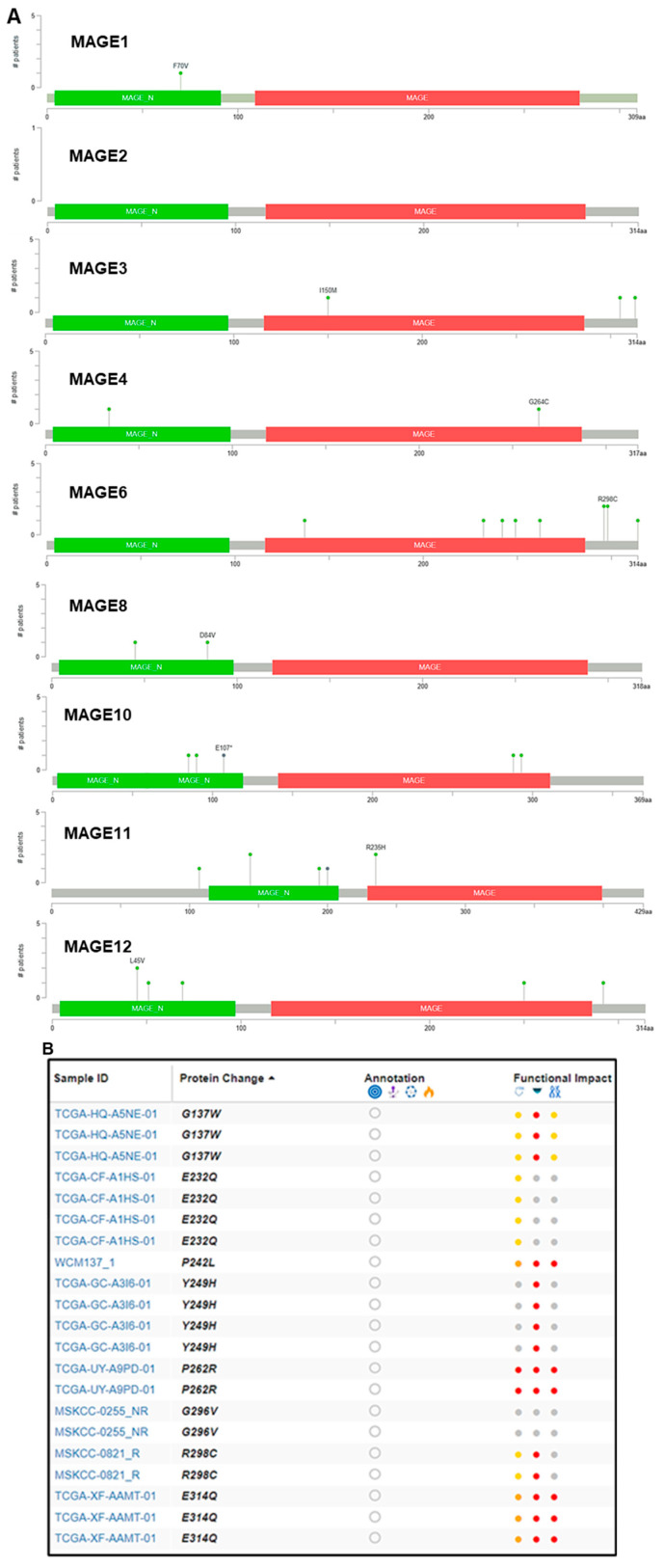
Mutation in the MAGEA family in bladder cancer. (**A**) Mutation plot was generated by the Mutation Mapper tool (cBioportal) showing the structure of MAGEA protein, and the frequency and position of mutations. Green color shows MAGE_N: Melanoma-associated antigen family N terminal (4–97), and red color shows MAGE: MAGE family (116–286). The green lollipop denotes number and change in the amino acid. (**B**) The table shows the patient’s TCGA sample ID, protein change, functional impact, and mutation type. Color dots denotes function impact: red—high, yellow/orange—moderate, and gray—low impact.

**Figure 5 cancers-16-00246-f005:**
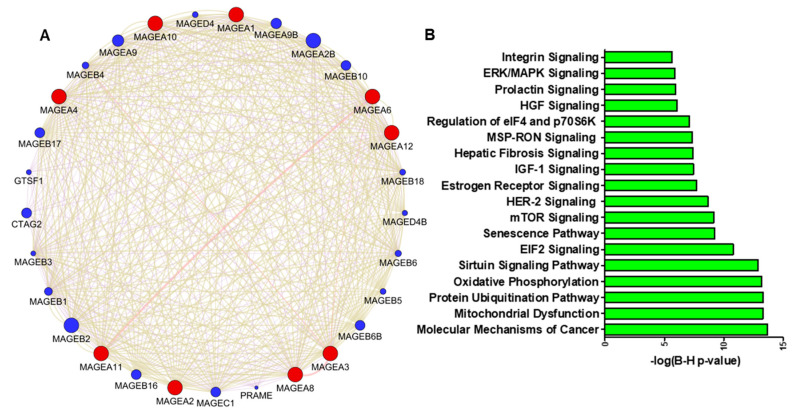
(**A**) Protein–protein interaction network of MAGEA family members constructed by Cytoscape software (version 3.10.1). Proteins are represented with color nodes, and interactions are represented with edges. The size of the circle of each protein represents its connection degree to other proteins. The red color circle shows the query protein, and the blue color shows the interacting proteins. (**B**) Signaling pathways associated with MAGEA family; the X-axis shows the log(B-H) value and the Y-axis represents the associated pathway in bladder cancer.

**Figure 6 cancers-16-00246-f006:**
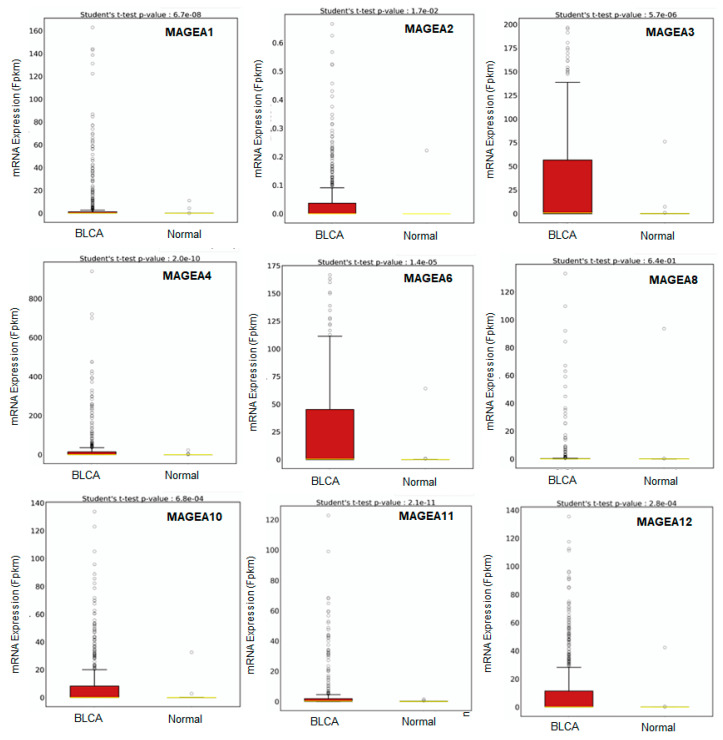
Transcriptomic expression of MAGEA family. Relative expressions of varying MAGEA types in normal samples and bladder cancer samples (BLCA) quantified through a box plot analysis of the TCGA dataset. The X-axis shows the expression of 435 patients (BCLA = 414, and Normal = 19), and the Y-axis shows the Fragments Per Kilobase of transcript per Million mapped reads (FPKM) mRNA expression values.

**Figure 7 cancers-16-00246-f007:**
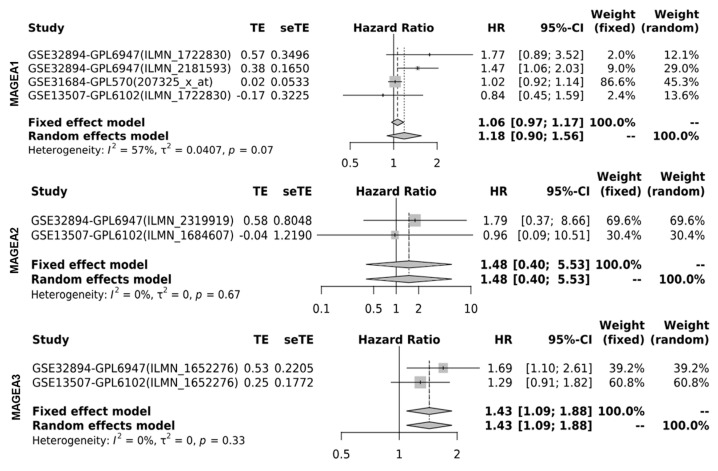
Hazard ratio (HR) of MAGEA proteins in bladder cancer. Forest plots were generated for hazard ratio in bladder cancer patients. HR was calculated using the fixed effect model and the random effect model with 95% CI with the *p*-value of heterogeneity.

**Figure 8 cancers-16-00246-f008:**
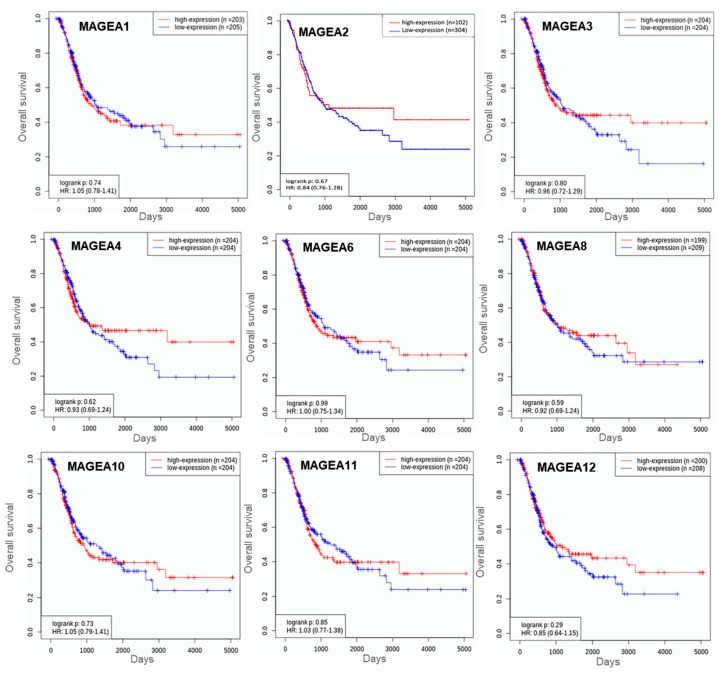
Kaplan–Meier plots for overall survival in bladder cancer. These curves were generated using the TCGA bladder cancer database of n = 408 patients. The X-axis shows the overall survival in days and the Y-axis shows the overall survival rate.

**Figure 9 cancers-16-00246-f009:**
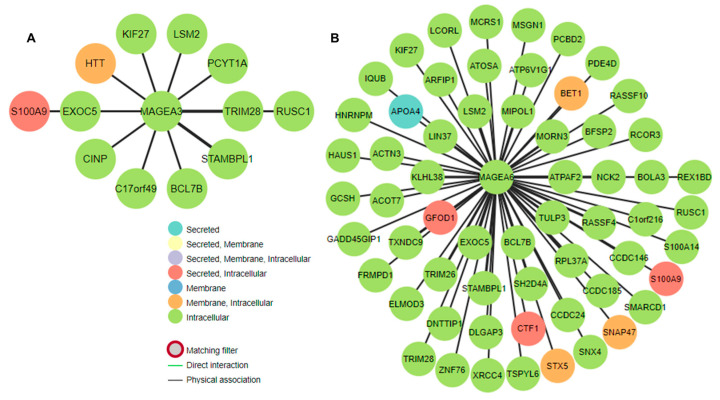
Gene interaction of (**A**) MAGEA3 and (**B**) MAGEA6 in bladder cancer. The figure shows the physical association and gene regulatory relationships. The assorted colors of the nodes show the predicted location of the proteins.

**Figure 10 cancers-16-00246-f010:**
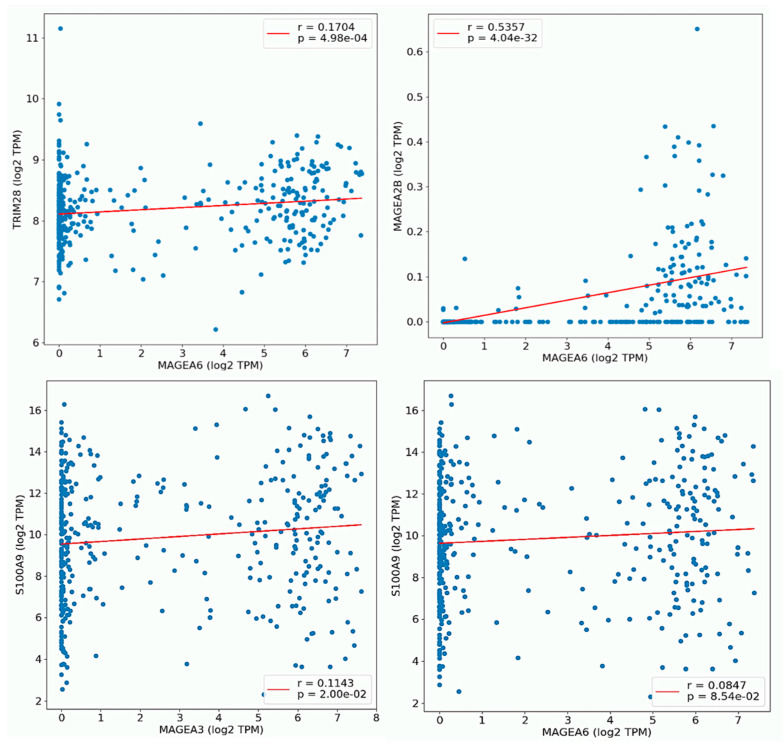
Plot pair-wise gene expression correlation analysis between MAGEA6, with TRM28, and MAGEA28. MAGEA3 showed its correlation with S100A9 in BLCA. The X and Y-axes of the graph show log2 transcript per million (TPM).

**Table 1 cancers-16-00246-t001:** Ongoing clinical trials of MAGEA family in bladder cancer.

SN	Clinical Trials	Trial Number	References
1	Safety and efficacy study of MAGE-A3 + AS-15 in patients with muscle-invasive bladder cancer after cystectomy	NCT01435356	[86]
2	Incidence of expression of tumor antigens in cancer tissue from patients with pathologically demonstrated bladder cancer.	NCT01706185	[61]
3	BCG modulation of the recMAGE-A3 + AS15 ASCI response in the treatment of non-muscle invasive bladder cancer (NMIBC) patients	NCT01498172	[87]
4	T cell receptor immunotherapy targeting MAGE-A3 for patients with metastatic cancer who are HLA-A*01 positive	NCT02153905	[88,89,90]
5	MAGE-A4^#1o32^T for multi-tumor	NCT03132922	[91]

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
