# Peer review of "Melanoma Antigen Family A (MAGE A) as Promising Biomarkers and Therapeutic Targets in Bladder Cancer"

_cancers, 2024, doi:10.3390/cancers16020246_

Round 1
Reviewer 1 Report
Comments and Suggestions for Authors
Authors performed a review regarding the prognostic role of MAGE proteins for bladder cancer as well as the potential to be therapeutic targets for a specific population of this disease. The subject is interesting, however the methodology is not clearly described in a dedicated section of the text rather than is scattered throughout the sections. In addition there are many technical considerations and difficult to interpret figures and graphs, which may be familiar for scientists with a genetic/molecular background but are not easily interpreted ny urologists/clinicians. I would suggest to make the text more clinically oriented, describe the methodology clearly in a dedicated section and improve the English language.
Comments on the Quality of English LanguageNeeds revision
Author Response
Comment. I would suggest making the text more clinically oriented, describe the methodology clearly in a dedicated section and improve the English language.
Response. We have used methods which are common and widely applied in the bioinformatics. The methods applied are discussed in each subsection. We feel that the methodology cannot be pooled together as it will provide more confusion to the readers. However, considering the comment, we have added method details in the beginning of each subsection to make the methodology easy to understand. We have also checked the manuscript for grammatical errors.
Reviewer 2 Report
Comments and Suggestions for Authors
The authors provide a thorough and detailed examination of the MAGEA family genes in bladder cancer, showcasing a commendable effort in collating and analyzing various studies. The inclusion of diverse aspects, such as gene expression, genomic alterations, signaling pathways, and therapeutic potential, adds depth to the review.
The use of large genomic databases like TCGA and GENT2 lends credibility and robustness to the findings. The extensive patient data analyzed helps in drawing more reliable and generalizable conclusions.
The exploration of MAGEA proteins as potential therapeutic targets, including the development of immunotherapies and small molecule inhibitors, represents a significant advancement and demonstrates the authors' innovative approach to cancer treatment.
Critical Comments:
1. Methodological Diversity Across Studies: The span of two decades across the studies reviewed may introduce variability in methodologies and diagnostic criteria. This variation could impact the consistency and comparability of results, and the authors might consider addressing these discrepancies more thoroughly.
2. Statistical Versus Clinical Significance: While statistical significance is often reported, the review could benefit from a deeper exploration of the clinical relevance of these findings. The authors are encouraged to provide more insight into how these statistically significant associations translate into clinical practice or patient outcomes.
3. Lack of Mechanistic Details: The review, while extensive in data, lacks in-depth discussion on the mechanistic pathways through which MAGEA genes influence tumor behavior. Elucidating these mechanisms could enhance the understanding of their role in bladder cancer and aid in the development of targeted therapies.
Points for Improvement:
1. Enhanced Focus on Mechanistic Pathways: Delve deeper into the biological mechanisms underlying the observed gene expressions and genomic alterations. This could include exploring how these genes interact with other molecular pathways and their direct impact on bladder cancer progression.
2. Contextualization with Other Genetic Alterations: Situate the findings within the broader landscape of bladder cancer genomics. Comparing MAGEA gene alterations with other known genetic changes in bladder cancer could provide a more comprehensive view of the disease’s molecular profile.
3. Translational Relevance: Expand the discussion on the translational aspect of these findings. How can the insights from this review be applied in clinical settings? This could involve examining the potential for developing new diagnostic markers, therapeutic targets, or treatment strategies based on MAGEA gene expressions and interactions.
Conclusion:
The manuscript presents a valuable and comprehensive exploration of MAGEA family genes in bladder cancer. While the breadth of data and innovative therapeutic insights are commendable, the review could be further strengthened by addressing methodological variations, enhancing the focus on mechanistic pathways, and expanding on the translational relevance of the findings.
Author Response
Comment. Methodological Diversity Across Studies: The span of two decades across the studies reviewed may introduce variability in methodologies and diagnostic criteria. This variation could impact the consistency and comparability of results, and the authors might consider addressing these discrepancies more thoroughly.
Response. The point is well taken. To address this issue, we have assembled data from the TCGA database and applied various statistical parameters for analysis and minimize random errors. One of the main advantages of this large database is not the robust sample size but to provide a more diverse patient source and comprehensive data analysis which can be used to discover various targets and pathways that contribute to the disproportionate bladder cancer burden and expand the potential of clinical research. The variation in the dataset was addressed through statistical analysis, in particular, we have used the Bonferroni test which is a multiple-comparison correction. Also, cancer-specific mortality was analyzed using hazard ratio (HR) with a fixed effect model and the random effect model with 95% CI. All the above statistical approaches were used to avoid any discrepancies in the dataset.
Comment. Statistical Versus Clinical Significance: While statistical significance is often reported, the review could benefit from a deeper exploration of the clinical relevance of these findings. The authors are encouraged to provide more insight into how these statistically significant associations translate into clinical practice or patient outcomes.
Response. We thank the reviewer for this encouraging comment. We have analyzed the statistical versus clinical significance mentioning approximately 47 clinical trials on MAGEA family, including 5 involving bladder cancer. Clinical trials focusing on MAGEA proteins in bladder cancer is summarized in Table 1. Moreover, we have also included clinical significance in requisite places in the text.
Comment. Lack of Mechanistic Details: The review, while extensive in data, lacks in-depth discussion on the mechanistic pathways through which MAGEA genes influence tumor behavior. Elucidating these mechanisms could enhance the understanding of their role in bladder cancer and aid in the development of targeted therapies.
Response. The point is well taken, and we have included additional mechanistic studies which causes aberrant MAGEA expression in various human cancers.
Points for Improvement
Comment. Enhanced Focus on Mechanistic Pathways: Delve deeper into the biological mechanisms underlying the observed gene expressions and genomic alterations. This could include exploring how these genes interact with other molecular pathways and their direct impact on bladder cancer progression.
Response. We have included additional mechanistic studies which causes alteration in MAGEA expression in various human cancers.
Comment. Contextualization with Other Genetic Alterations: Situate the findings within the broader landscape of bladder cancer genomics. Comparing MAGEA gene alterations with other known genetic changes in bladder cancer could provide a more comprehensive view of the disease’s molecular profile.
Response. We have included genetic changes which are responsible for aberrant MAGEA expression in bladder cancer including alteration TP53 (50.88%, Non-papillary, 30.65% invasive, 61.36% non-papillary) is on the top gene list with high genetic alteration in bladder cancer followed by PIK3CA (Non-papillary), HRAS (Invasive) and several others highlighted in the manuscript.
Comment. Translational Relevance: Expand the discussion on the translational aspect of these findings. How can the insights from this review be applied in clinical settings? This could involve examining the potential for developing new diagnostic markers, therapeutic targets, or treatment strategies based on MAGEA gene expressions and interactions.
Response. The translational relevance of these findings is discussed in therapeutic target subsection.
Reviewer 3 Report
Comments and Suggestions for Authors
This study summarizes the significance of MAGE A.
This information is beneficial for bladder cancer patients.
Author Response
Comment. This study summarizes the significance of MAGE A. This information is beneficial for bladder cancer patients.
Response. We appreciate the reviewer’s comments on the manuscript. There are no concerns to address.
Round 2
Reviewer 1 Report
Comments and Suggestions for Authors
authors responded
Reviewer 2 Report
Comments and Suggestions for Authors
Thank you for addressing the concerns raised in the previous review.
The addition of TCGA database data and the application of various statistical parameters, including the Bonferroni test, enhance the robustness of your study. It's commendable how these changes help in minimizing potential discrepancies and improving the comprehensiveness of your analysis.
The inclusion of information on clinical trials related to MAGEA family proteins in bladder cancer, especially in the context of statistical versus clinical significance, is a valuable addition. It helps in bridging the gap between research findings and clinical application.
Your effort to include mechanistic details of the MAGEA genes in tumor behavior is appreciated. This not only addresses a key concern but also enriches the study by providing insights into potential therapeutic targets.
The contextualization of MAGEA gene alterations within the broader genomic landscape of bladder cancer and the discussion on translational relevance are thoughtful enhancements. These additions provide a more comprehensive understanding of the disease and its treatment possibilities.
Overall, your revisions have meaningfully addressed the initial comments and improved the manuscript. Your efforts to refine the study based on feedback are evident and appreciated.
Best regards,
[Reviewer's Name]